# Spatial heterogeneity of microbial community structure and its environmental drivers in surface sediments of Erhai Lake

Hong Xiang[1☯], Cong-Gao Yang[2☯], Hao-Qin Xiong[1], Hao-Ran Bao[1], Jia-Zhuo Qu[1], Zhe-Xi Luan[1], Xiao-Long Sun [1]*, Li Zhen[1]*

**1** Yunnan Key Laboratory of Plateau Wetland Conservation, Restoration and Ecological Services, College of Ecology and Environment, Southwest Forestry University, Kunming, China, **2** Kunming Appraisal Center for Ecological Environment and Engineering, Kunming, China

☯ These authors contributed equally to this work.
* sunxl96@gmail.com (X-LS); zl@swfu.edu.cn (LZ)

## Abstract

As a crucial plateau freshwater lake in Yunnan Province, China, Erhai Lake exhibits distinct environmental heterogeneity driven by its unique watershed characteristics and human activities, significantly influencing sediment microbial communities. This study investigated the spatial relationships between environmental factors and microbial community structures in surface sediments from the eastern, western, and northern shores using redundancy analysis (RDA) and Spearman correlation analysis. Results revealed that pH, total nitrogen (TN), total phosphorus (TP), total organic carbon (TOC), and redox potential (Eh) were key drivers of microbial community divergence. The western shore, with the highest TP, TOC, and nitrogen levels, displayed elevated microbial diversity dominated by *Proteobacteria* and *Bacteroidetes*, reflecting heterotrophic adaptations to elevated pollution loads. The northern shore exhibited severe nitrogen pollution, marked by the highest TN content and enrichment of *Thiobacillus* sp., potentially enhancing water self-purification. The eastern shore, with minimal anthropogenic disturbance, showed the highest bacterial diversity but the lowest nutrient concentrations. Fungal community structure was significantly influenced by pH, Eh, and TOC, while ecological restoration measures on the western shore enhanced fungal community stability. This study highlights how spatial heterogeneity in environmental factors regulates microbial community structure and function, ultimately affecting the stability of lake ecosystems. These findings provide a scientific basis for ecological restoration and sustainable management of plateau lakes.

## Introduction

Erhai Lake is an important freshwater lake in Yunnan Province, China, with significant ecological, economic, and cultural values. However, due to increased human

**Data availability statement:** within the manuscript and/or Supporting information files

**Funding:** This study was sponsored by the Yunnan Province Agricultural Basic Research Joint Special Project (202301BD070001-065), the Key R&D Plan Projects in Yunnan Province(202203AC100002-03) and the Scientific Research Fund of Yunnan Provincial Education Department, China (2022J0524).

**Competing interests:** The authors have declared that no competing interests exist.

activities and the impact of climate change, the water quality and ecosystem of Erhai Lake have changed significantly [1–3]. Previous research has shown that there are obvious environmental differences between different shore areas of Erhai Lake, and these differences have a profound impact on the structure and function of the lake's ecosystem [4].

Microbial communities play a key role in lake ecosystems, participating in essential processes such as material circulation, energy flow, and pollutant degradation [5,6]. Especially in plateau lakes like Erhai Lake, fungi play an important role in sediments. As the main decomposers of plant matter, they can break down highly resistant lignocellulosic fibers and play an irreplaceable role in maintaining the balance of the ecosystem [7,8]. The diversity and distribution of these microbial communities are significantly affected by various environmental factors, such as pH, nutrients, and human activities (such as agricultural pollution and ecological restoration projects), which in turn affect the structure and function of microbial communities [9–11]. Nutrients in sediments can be easily released into the water body under certain conditions, thereby affecting the water body's nutritional status and ecological balance [12]. Agricultural activities and ecological restoration measures also intensify these effects, resulting in significant differences in the diversity and structure of microbial communities in different regions. Erhai Lake, a typical fault depression lake basin, has a shorter river system with the north shore being the longest, followed by the west shore, and the east shore being the shortest. Longer river systems lead to more frequent agricultural activities. Therefore, it is necessary to study how these environmental factors affect the structure of microbial communities in plateau lake sediments, which is important for understanding and protecting lake ecosystems.

This study aims to deeply explore the spatial distribution characteristics of microbial communities in the surface sediments of Erhai Lake and their relationship with environmental factors. Especially in different areas on the east, west, and north shores of the Erhai Basin, the structure and function of microbial communities may show significant spatial heterogeneity due to the impact of agricultural activities and social life. This study systematically analyzed the correlation between microbial community structure and environmental factors in different shore areas of Erhai Lake through Bray-Curtis distance and NMDS analysis, redundancy analysis (RDA), and Spearman correlation analysis. The results of this study not only deepen the understanding of the Erhai sediment microbial community structure and its environmental driving factors but also provide a scientific basis for the protection and sustainable management of plateau lake ecosystems.

## Materials and methods

### Sample collection

Taking the Erhai Lake Basin as the research object, surface sediments along the shore of the basin were collected for processing and analysis. The water surface area of Erhai Lake is 250km$^2$, with a lake capacity of 2.88 billion m$^3$ and an average depth of 10.2m. It is supplied by rainfall and spring water in the basin and flows into Erhai Lake through 117 rivers (streams). The Miju River in the north is the main

source of Erhai Lake, with a length of about 21 km. There is also the Cangshan 18 Creek in the west, with a length of 8–12 km, and the Haichao River, Fengwei Creek, and Yulong River in the east. The rivers are mostly straight and short, with steep riverbed slopes. Sampling was conducted in June 2020 during the early wet season, a period of stable hydrological conditions in Erhai Lake. This timing minimizes short-term fluctuations and prioritizes spatial heterogeneity analysis [2]. 16 sampling points were set up on the shoreline near the estuaries of rivers along Erhai Lake (east shore: Xiahe Village (XHC), Langqiao (LQ), Lianhuaqu (LHQ), Kanglang Village (KLC); west shore: Yiheyuan (YHY), Taihe Village (THC), Dazhuang (DZ), Xiaoyizhuang (XYZ), Xiajiyi (XJY), Fumeiyi (FMY), Beipan (BP), Renliyi Village (RLYC), Taoyuan (TY); north shore: Xia Village (XC), Shaping Village (SPC), Yangjia Village (YJC)), as shown in Fig 1. Sediment samples (0–10 cm depth) were collected from designated sampling points on the eastern, northern, and western shores of Erhai Lake using a grab sampler. Samples from each point were transferred to a clean plastic bucket and thoroughly homogenized by hand (using gloved hands) to eliminate vertical stratification effects. From each homogenized composite, two

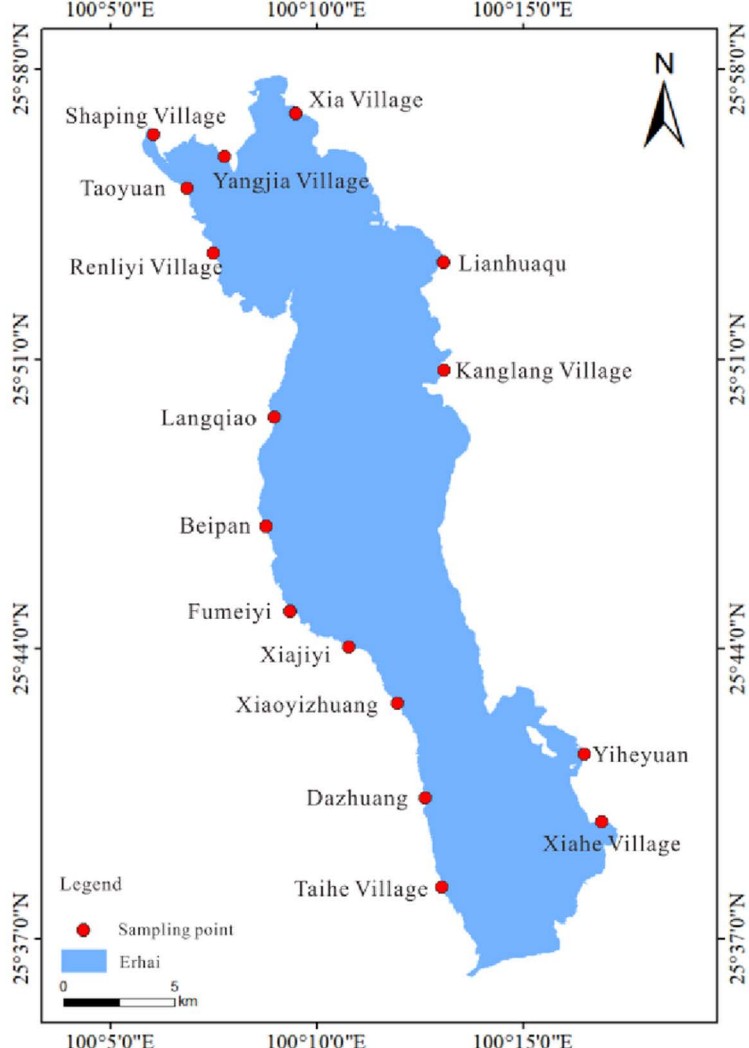

**Fig 1. Sampling point map.** (Base geographic data from the National Platform for Common Geospatial Information Service (Tianditu, map.tianditu.gov.cn) under the Open Government Data License of China).

subsamples were allocated, one for physicochemical analysis and another for microbiological analysis. Both subsamples were placed in separate ziplock bags labeled with their respective sampling point codes and immediately stored in a low-temperature incubator. Triplicate samples were collected per sampling point. Following collection, subsamples designated for microbiological analysis were transported to an accredited testing facility. The remaining subsamples were air-dried at ambient temperature in a ventilated environment. During drying, macroscopic debris (plant matter, animal remains, and gravel) was manually removed. Processed samples were then sealed in airtight bags for subsequent physicochemical characterization.

### Determination of physical and chemical properties of sediment

All indicators were brought back to the laboratory for measurement. The pH value of the sediment was measured using a pH meter, the moisture content (MC) was determined by weighing method, the total organic carbon (TOC) was measured using an organic carbon analyzer (German Element Vario), and the total nitrogen (TN), total phosphorus (TP), ammonia nitrogen ($NH_3$-N), and nitrate nitrogen ($NO_3^-$-N) were measured using a continuous flow analyzer (German SEAL Analytical AA3).

### Microbial sequencing

The bacterial and fungal communities in the sediments were determined by 16S/ITS rRNA amplicon sequencing. After sampling, Guangdong Meige Gene Technology Co., Ltd. was commissioned to analyze and determine the community diversity and composition of bacteria and fungi in the sediments.

### Statistical analysis

The α-diversity index was used to analyze the microbial structure and evaluate the diversity level of microorganisms in each shore area. The β-diversity index was used to compare the differences and similarities of microbial communities between different shore areas, revealing the spatial distribution characteristics of microbial communities under different environmental conditions. To analyze the similarities and differences in microbial community structure in sediments from different sampling areas, the multidimensional data were analyzed by dimensionality reduction using NMDS under the Bray-Curtis distance. The vegan package of R was used for redundancy analysis (RDA) to study the relationship between environmental parameters and bacterial community structure, revealing which environmental factors may be important in driving differences in dominant microbial communities. The similarity analysis (Anosim) in the vegan package was used to determine the effects of environmental conditions in different shore areas on bacterial communities. Finally, Spearman correlation analysis was used to further explore the association between specific microbial taxa and environmental factors and to verify the specific mechanisms by which environmental factors influence the main bacterial species.

## Results and analysis

### Physical and chemical properties of sediments in the study area

As shown in Table 1, The pH of sediments in Erhai Lake is generally alkaline, with the east shore ranging from 8.03 to 8.62, the north shore from 7.57 to 8.24, and the west shore from 7.19 to 7.95. The east shore has the highest pH, and the west shore has the lowest. The average TN concentration is 2.239 g/kg, with the lowest on the east shore and the highest on the north shore. The average TP concentration is 0.941 g/kg, with the lowest on the east shore and the highest on the west shore. The average TOC concentration is 19.766 g/kg, lowest on the east shore and highest on the west shore. $NH_3$-N averages 0.139 mg/kg, lowest on the east shore and highest on the west shore. $NO_3^-$-N averages 0.035 mg/kg, also lowest on the east shore and highest on the west shore. Overall, Erhai Lake's nitrogen and phosphorus pollution is serious, with significant phosphorus release risk and high organic matter concentrations promoting nitrogen salt release. The pattern for TP and TOC is west shore > north shore > east shore, and for TN, it's north shore > west shore > east shore.

**Table 1. Physical and chemical properties of each sampling point.**

| Sampling sites | | pH | TOC (g/kg) | NH$_3$-N (mg/kg) | NO$_3^-$-N (mg/kg) | MC (%) | DOC (mg/L) | TN (g/kg) | TP (g/kg) | Eh (mv) |
|---|---|---|---|---|---|---|---|---|---|---|
| East shore | XHC | 8.6 | 10 | 0.0052 | 0.015 | 29 | 17 | 0.12 | 0.14 | −84 |
| | YHY | 7.9 | 19 | 0.024 | 0.017 | 43 | 16 | 0.72 | 0.66 | −42 |
| | LHQ | 8.0 | 14 | 0.020 | 0.013 | 37 | 17 | 3.08 | 0.60 | −50 |
| | KLC | 7.9 | 21 | 0.090 | 0.028 | 54 | 18 | 3.0 | 2.3 | −44 |
| north shore | SPC | 7.5 | 33 | 0.071 | 0.029 | 61 | 16 | 3.9 | 1.0 | −17 |
| | YJC | 8.2 | 12 | 0.0093 | 0.012 | 32 | 15 | 1.9 | 0.98 | −62 |
| | XC | 8.2 | 14 | 0.039 | 0.017 | 48 | 17 | 2.1 | 0.83 | −60 |
| west shore | THC | 7.2 | 31 | 0.39 | 0.042 | 74 | 18 | 1.1 | 0.40 | −18 |
| | DZ | 8.0 | 11 | 0.014 | 0.013 | 25 | 16 | 0.27 | 0.62 | −50 |
| | XYZ | 7.8 | 10 | 0.0077 | 0.014 | 42 | 16 | 5.1 | 0.95 | −37 |
| | XJY | 7.8 | 12 | 0.025 | 0.017 | 46 | 16 | 1.2 | 0.63 | −36 |
| | FMY | 8.3 | 10 | 0.0052 | 0.011 | 31 | 15 | 1.0 | 1.6 | −64 |
| | BP | 7.4 | 12 | 0.037 | 0.014 | 36 | 15 | 1.8 | 1.4 | −38 |
| | LQ | 7.7 | 18 | 0.10 | 0.016 | 61 | 17 | 2.8 | 1.1 | −30 |
| | RLYC | 7.7 | 13 | 0.054 | 0.022 | 35 | 16 | 1.7 | 1.1 | −31 |
| | TY | 7.4 | 76 | 1.3 | 0.28 | 86 | 17 | 6.5 | 1.0 | −16 |

## Microbial community structure in sediments

To further explore the distribution characteristics of microorganisms in the sediments of the Erhai Lake Basin, the composition of microbial communities was analyzed at the phylum and genus levels.

### Analysis of dominant bacterial communities at the phylum level

At the phylum level, microorganisms with a relative abundance of bacteria and fungi ≥ 10% were plotted as bar graphs (Fig 2). The results showed that the bacterial community was relatively rich, with consistent trends but varying relative abundances across samples. *Proteobacteria* was the most abundant phylum, followed by *Bacteroidetes*, *Chloroflexi*, *Acidobacteria*, and *Actinobacteria*. The fungal community also showed richness and consistency in trends, with varying relative abundances. *Ciliophora* was the most abundant, followed by *Ascomycota*, *Rozellomycota*, *Chytridiomycota*, and *Basidiomycota*.

### Analysis of dominant bacterial communities at the genus level

The bacterial community structure at the genus level in sediments from different sampling points is shown in Fig 3a. The results show that the bacterial richness of the surface sediments of Erhai Lake is high, which is consistent with the results of α-diversity analysis.Dominant bacterial genera include *Crenothrix* sp. (west shore > east shore), *Ellin6067* sp.(north shore > west shore > east shore), *Thiobacillus* sp. (north shore > east shore > west shore), *Luteolibacter* sp. (west shore > east shore > west shore, the west shore is absolutely dominant), Sva0081_sediment_group (west shore > east shore > north shore, the west shore is absolutely dominant), *Spirochaeta* sp. (east shore > north shore > west shore), etc.

At the fungal genus level, shown in Fig 3b, the dominant fungal genera include *Frontonia* sp. (west shore > east shore > north shore, with the west shore being absolutely dominant), *Prorodon* sp. (west shore > north shore > east shore), *Arcuospathidium* sp. (north shore > east shore > west shore), *Emericellopsis* sp. (east shore > north shore > west shore), *Cyrtohymena* sp. (north shore > east shore > west shore), *Vorticellides* sp. (west shore > east shore > north shore), etc.

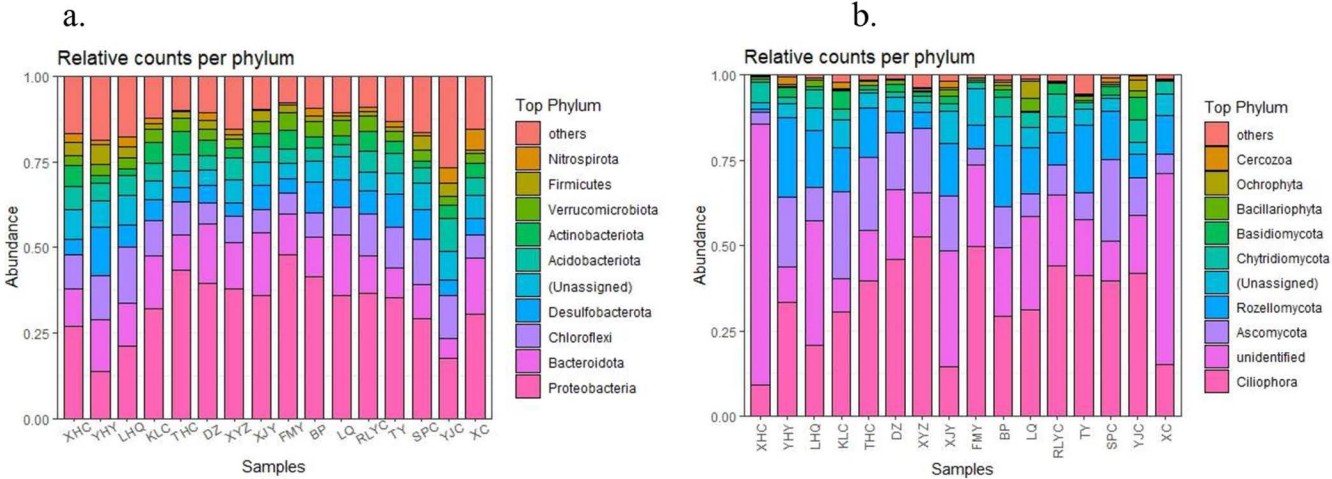

**Fig 2. Community structure of bacteria (a) and fungi (b) in water from different sampling points.**

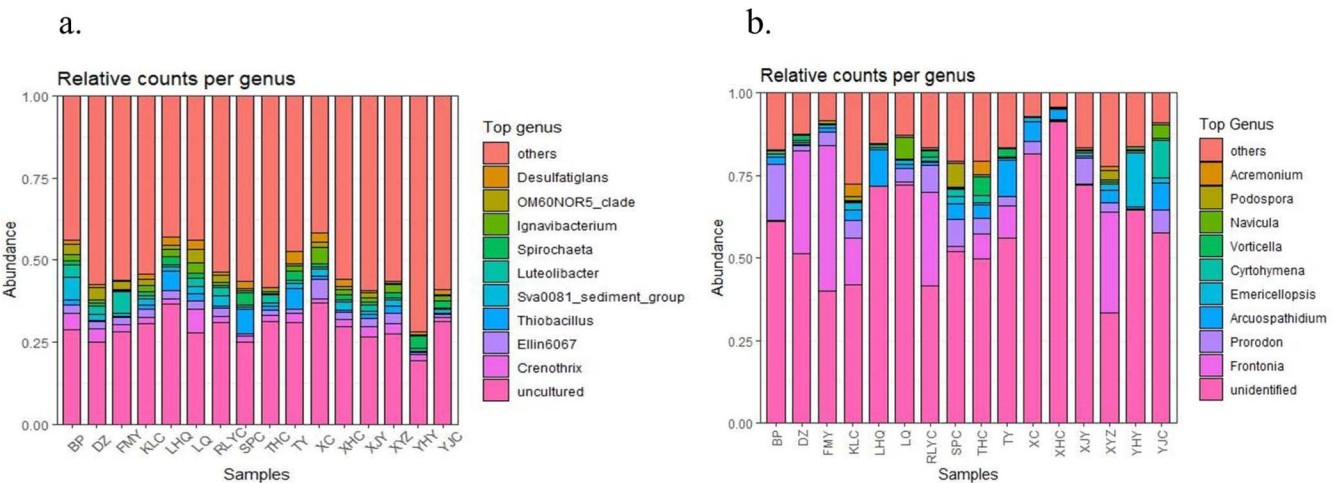

**Fig 3. Bacterial (a) and fungal (b) community structures at the genus level at different sampling points.**

## Analysis of sediment microbial diversity and community structure

**Analysis of microbial α diversity.** In the sediment samples of the Erhai Lake Basin, rich bacterial and fungal communities were detected, with a total of 40,387 bacterial OTUs and 7,221 fungal OTUs obtained, as shown in Fig 4. There were 69 phyla, 141 classes, 296 orders, 383 families, and 678 genera of bacteria, and 37 phyla, 76 classes, 156 orders, 383 families, and 431 genera of fungi annotated. Bacterial diversity analysis showed that the average values of the Shannon index from low to high were west shore (8.033), north shore (8.1), and east shore (8.16). The average values of the Simpson index were east shore (0.000745) <north shore (0.00101) <west shore (0.00114), and the Chao index was west shore (8668.667) <east shore (8837.75) <north shore (9139). In terms of fungal diversity, the average Shannon index was north shore (4.4067) <east shore (4.6775) <west shore (4.91). The average Simpson index was west shore (0.037) <east shore (0.0386) <north shore (0.0636), and the Chao index was north shore (521.67) <east shore (615) <west shore (768.22).

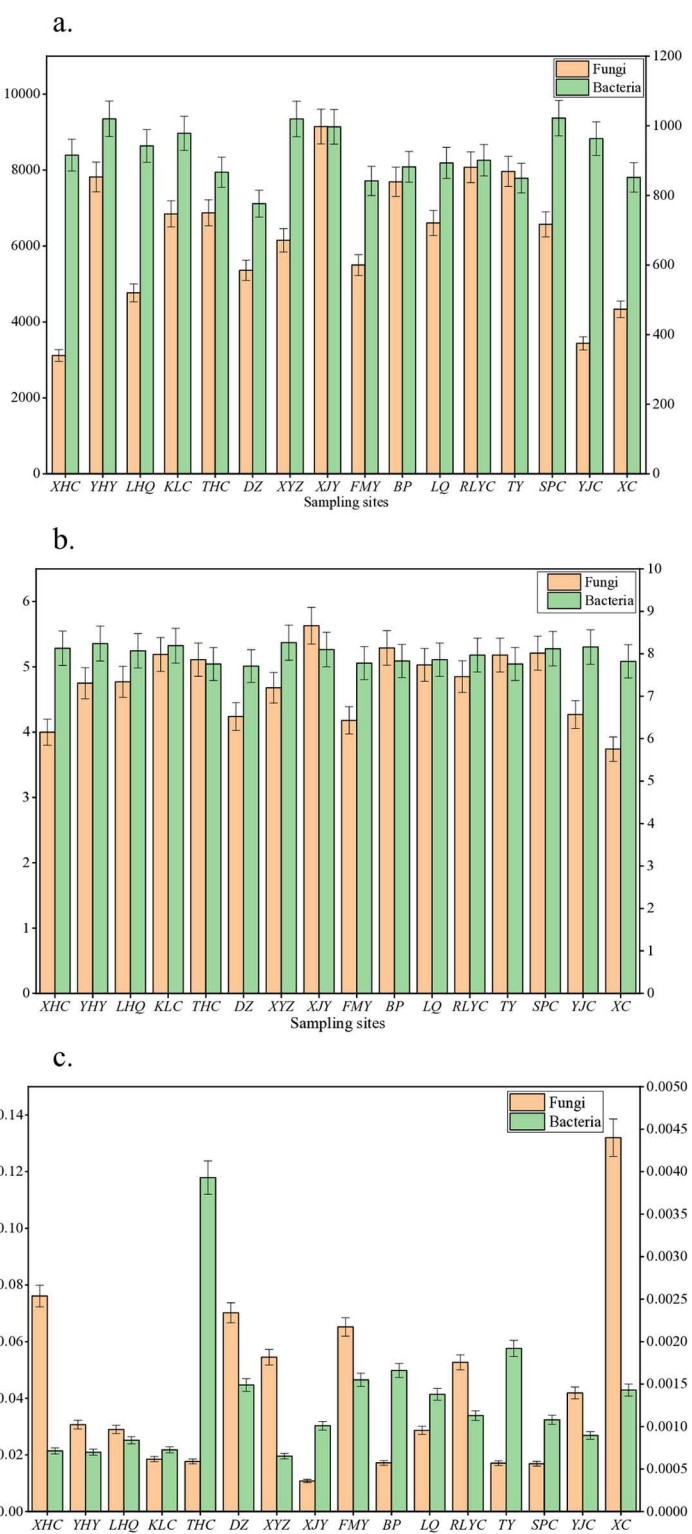

**Fig 4. Alpha diversity bar graph of microbial communities in surface sediments of Erhai Lake.** (a. Chao index; b. Shannon index; c. Simpson index).

**Microbial β-diversity analysis.** This study compared bacterial and fungal community structures at different Erhai Lake sampling sites using the beta diversity index. The OTU-level species differences are displayed in a sample correlation distance heat map. Darker colors indicate greater species diversity between sampling points. Fig 5 shows significant differences in bacterial and fungal community structures across sampling points.

As can be seen from Fig 5a, the distance coefficients of the sampling points on the west shore are generally small, especially the distance coefficients of the sampling points LQ and BP which is 0.643, indicating that the bacterial communities of Langqiao and Beipan have a high degree of similarity. The distance coefficients between YHY on the east shore and multiple sampling points are relatively large, with the distance coefficients of XC and YJC on the north shore and XYZ and XHC on the west shore being 0.946, 0.944, 0.943, and 0.940, respectively. This indicates that the sediment bacterial community of the Yiheyuan sampling point is quite different from that of other sampling points and has specificity. These differences align with higher TN, TP, TOC, $NH_3$-N, and $NO_3^-$-N contents on the west and north shores compared to the east shore.

As can be seen from Fig 5b, the distance coefficients of the sampling points on the west shore are generally small, especially the distance coefficients of the THC and SPC sampling points, which is 0.061, indicating that the fungal communities of Taihe Village and Shaping Village have a high degree of similarity. The distance coefficients between XHC on the east shore and multiple sampling points are relatively large, among which the distance coefficients with SPC on the north shore, YHY, XYZ, and THC on the west shore are 0.710, 0.717, 0.683, and 0.672, respectively. This indicates that the sediment fungal community at the Xiahe Village sampling point is quite different from that at other sampling points and has specificity. These results highlight significant differences in sediment fungal community structures between the east, west, and north shores.

To visualize the differences in community composition between samples, NMDS plots were calculated based on the Bray-Curtis dissimilarity index as a measure of beta diversity. The relevant results are shown in Fig 6. It can be seen from Fig 6a and 6e that the sampling points on the east shore and the north shore are relatively scattered, while the sampling points on the west shore are relatively concentrated. ANOSIM was used to further determine the similarity between the west shore,

a.                                              b.

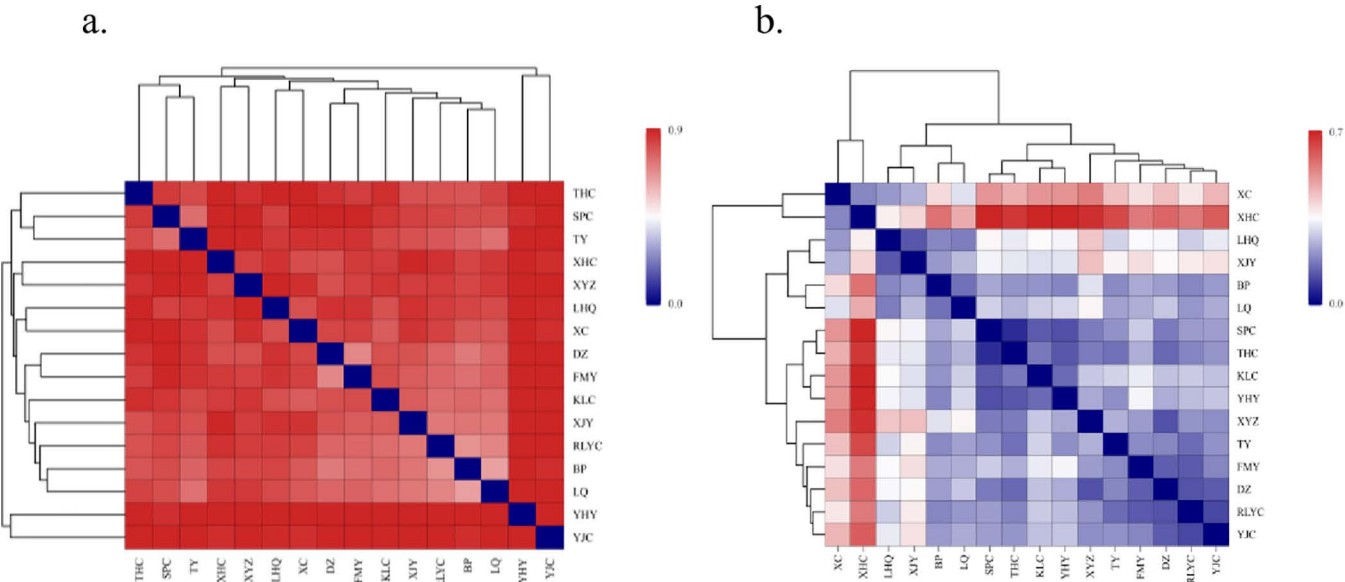

**Fig 5. Heatmap of sample distances at different sampling points (a. Bacteria, b. Fungi).**

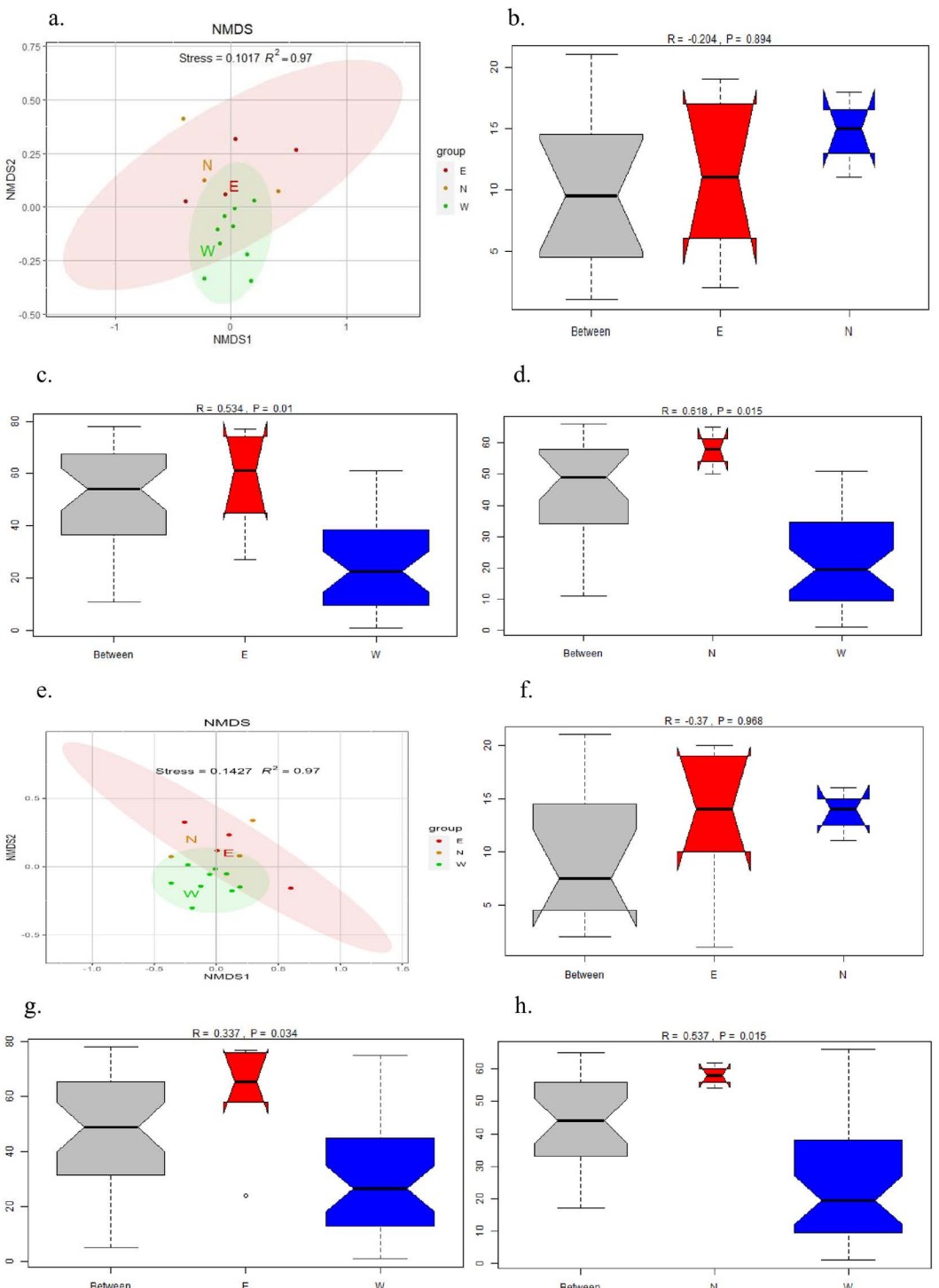

**Fig 6. Bacterial community composition visualized by NMDS ordination (Bray-Curtis distance); e. Fungal community composition visualized by NMDS ordination; b-d: intergroup differences in bacteria; f-h: intergroup differences in fungi; (E: east shore, W: west shore, N: north shore).**

the east shore, and the north shore. The results showed that there was no significant difference in the microbial community structure between the east shore and the north shore (Fig 6b and 6f). However, there were significant differences in the microbial community structure between the east shore, the north shore, and the west shore (Fig 6c, 6d, and 6g, 6h).

As shown in Fig 7, the interaction between bacterial and fungal diversity, richness, and environmental factors in Erhai Lake was determined using the Spearman rank correlation coefficient. There is no significant correlation between bacterial alpha diversity (Simpson, Chao1, and Shannon indices) and environmental factors. However, fungal alpha diversity shows a significant correlation with environmental factors such as pH and Eh (Spearman rank correlation analysis, $P < 0.05$).

## Correlation between microbial community structure and environmental factors

**Redundancy analysis.** The structure of microbial communities is intricately linked to environmental factors. Understanding their impact on microbial diversity is crucial. At the bacterial genus level, a redundancy analysis (RDA) diagram (Fig 8a) showed that across samples from the east shore, west shore, and north shore of Erhai Lake, the first two axes of RDA explained 38.3% and 30.8% of the variation, respectively, totaling 69.1%. TN, MC, and TOC were key positive factors on the first axis, while pH and TP were significant negative factors. The second axis was positively influenced by pH and negatively by TOC, highlighting the influence of sediment MC, TN, and TOC on microbial community structure.

At the fungal genus level, a redundancy analysis diagram (Fig 8b) demonstrated that across samples from the east shore, west shore, and north shore of Erhai Lake, the first two axes of CCA explained 15.8% and 13.9% of the variance, totaling 29.7%. pH was the primary positive factor on the first axis, countered by Eh and MC as negative factors. On the second axis, sediment MC and DOC were influential positive factors, whereas pH played a dominant negative role. These results underscore the collective influence of sediment MC, Eh, pH, and DOC on shaping fungal community structure.

**Correlation analysis.** Redundancy analysis can only reflect the relationship between environmental factors and microbial genera in general. To clarify the correlation between microbial genera and environmental factors in the surface

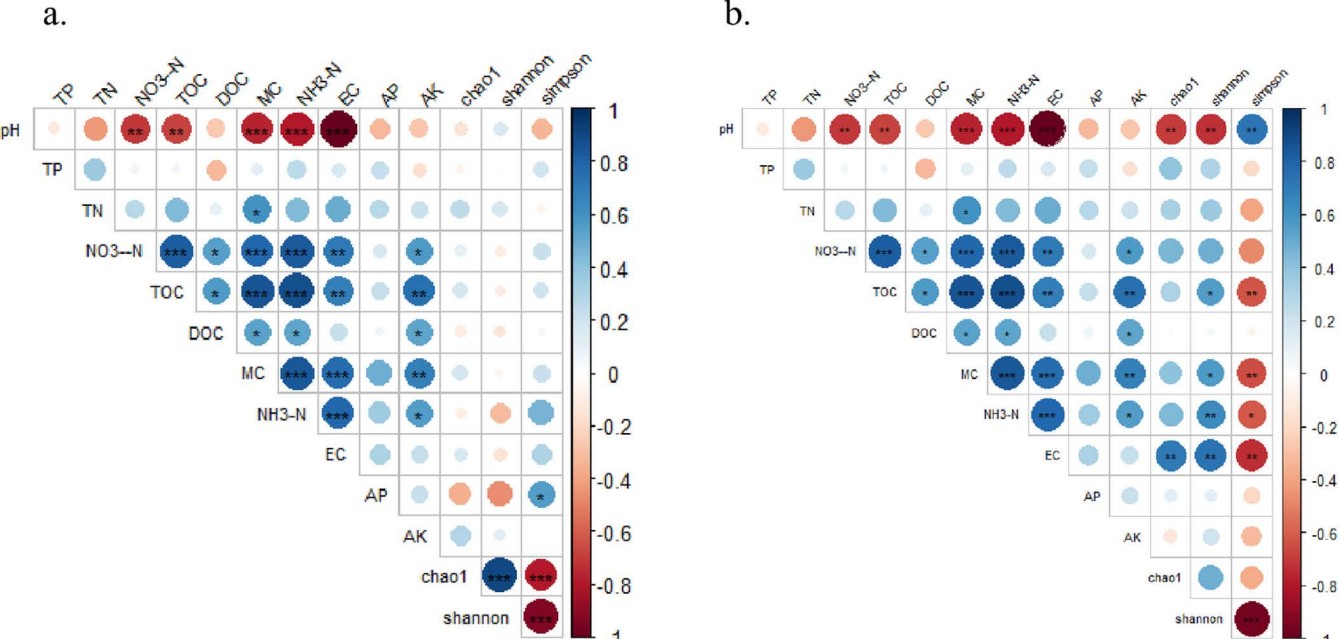

**Fig 7. Interactions between diversity, richness and environmental factors of bacteria (a) and fungi (b) in surface sediments of Erhai Lake.**

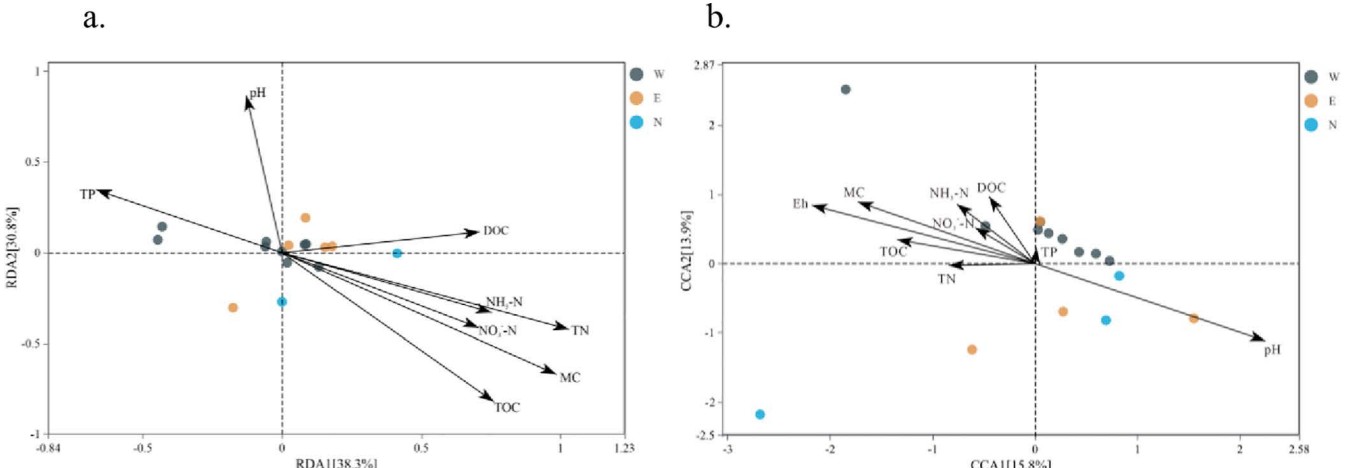

**Fig 8. Redundancy analysis of bacteria (a) and fungi (b) at the genus level and sediment physical and chemical properties (W: West Bank E: East Bank N: North Bank).**

sediments of Erhai Lake, Spearman correlation analysis was performed on the bacterial and fungal genus levels and environmental factors. The results are shown in Fig 9.

Seven bacteria (*Terrimonas* sp., *Lacibacter* sp., *Truepera* sp., *Meiothermus* sp., mle1−7, JGI_0001001-H03, and *Calothrix* sp.*KVSF5*) showed a significant negative correlation with TOC ($P < 0.01$), while three bacteria (*Thermoanaerobaculum* sp., ADurb.Bin063−1 and *Syntrophorhabdus* sp.) had a significant positive correlation with TOC. Five bacteria (*Terrisporobacter* sp., *Lacibacter* sp., *Truepera* sp., *mle1−7*, and *Meiothermus* sp.) displayed a significant positive correlation with pH, whereas three bacteria (*Methyloversatilis* sp., *Geobacter* sp., and *Syntrophorhabdus* sp.) had a significant negative correlation with pH. For Eh, the same five bacteria showed a significant negative correlation, while *Methyloversatilis* sp., *Geobacter* sp., and *Syntrophorhabdus* sp. had a significant positive correlation. Regarding NH$_3$-N, five bacteria (*Terrisporobacter* sp., *Lacibacter* sp., *Truepera* sp., *Meiothermus* sp., and JGI_0001001-H03) showed a significant negative correlation, whereas Subgroup_23, BD1−7_clade, and ADurb.Bin063−1 displayed a significant positive correlation. Lastly, six bacteria (*Sulfurifustis* sp., *Pleurocapsa* sp.PCC-7319, *Lacibacter* sp., *Truepera* sp., *Meiothermus* sp., and JGI_0001001-H03) had a significant negative correlation with MC, while *Thermoanaerobaculum* sp., ADurb.Bin063−1, and *Syntrophorhabdus* sp. exhibited a significant positive correlation with MC.

Six fungal genera (*Gibellulopsis* sp., *Gomphonema* sp., *Cytospora* sp., *Tintinnidium* sp., *Gaertneriomyces* sp. and *Trichoderma* sp.) showed a significant negative correlation with pH ($P < 0.01$). Seven genera (*Gibellulopsis* sp., *Gomphonema* sp., *Cytospora* sp., *Paramecium* sp., *Tintinnidium* sp., *Gaertneriomyces* sp. and *Trichoderma* sp.) were significantly positively correlated with Eh ($P < 0.01$). Five genera (*Askenasia* sp., *Fusarium* sp., *Gomphonema* sp., *Tintinnidium* sp. and *Gibellulopsis* sp.) exhibited a significant positive correlation with NH$_3$-N ($P < 0.01$). *Cytospora* sp., *Gomphonema* sp., *Cladosporium* sp. and *Leptodiscella* sp. displayed a significant positive correlation with TOC ($P < 0.05$). Three genera (*Fusarium* sp., *Leptodiscella* sp. and *Cytospora* sp.) were significantly positively correlated with NO$_3^-$-N ($P < 0.05$). *Cytospora* sp.and *Gomphonema* sp.also showed a significant positive correlation with MC ($P < 0.05$). Similar to bacteria, TN, TP, and DOC had little impact on the structure of fungal communities in sediments.

In summary, pH, Eh, NH$_3$-N, MC, and TOC are key factors affecting the structure of bacterial and fungal communities, while total nitrogen (TN), total phosphorus (TP), and dissolved organic carbon (DOC) have little effect on the structure of microbial communities.

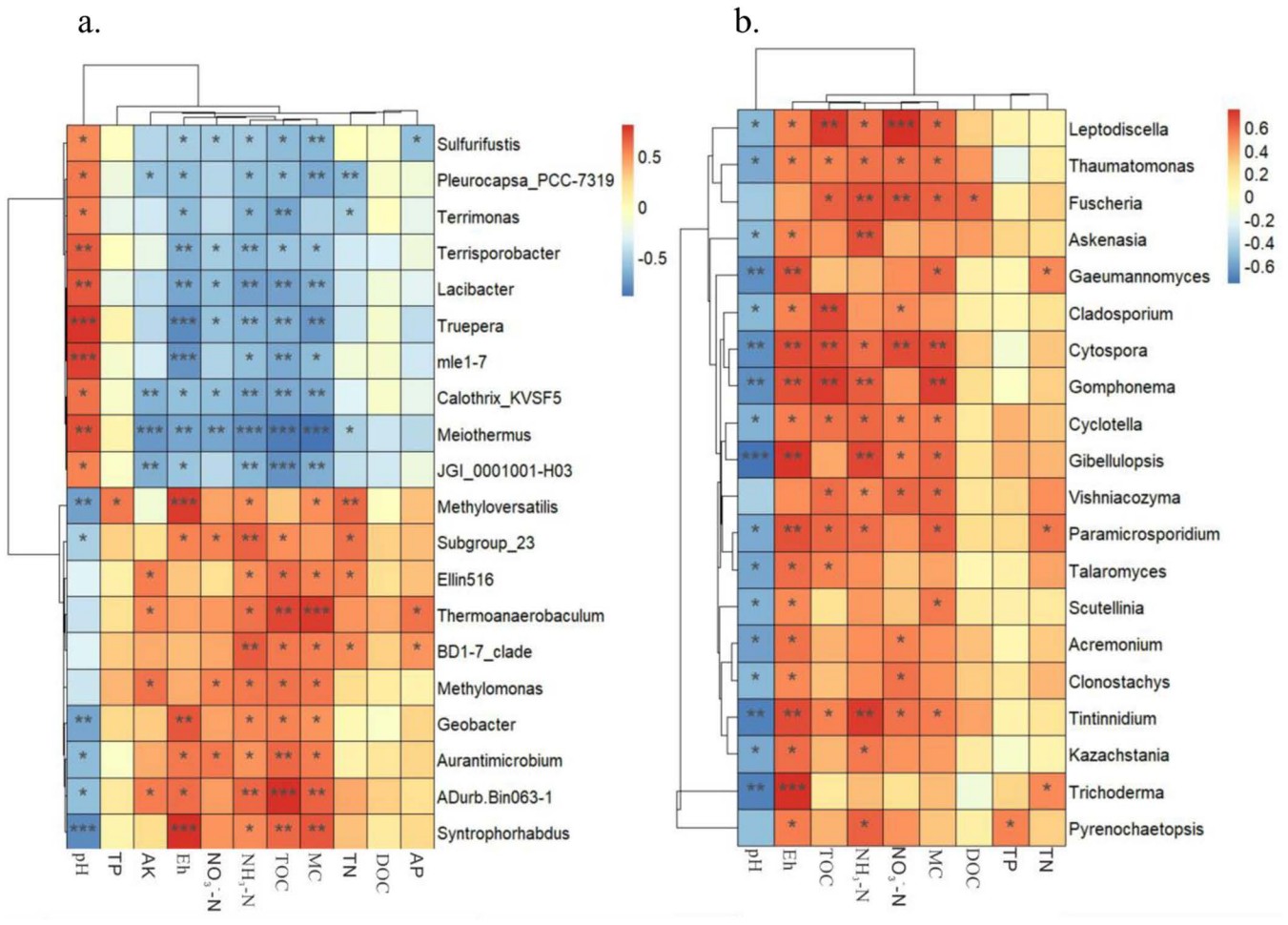

**Fig 9. Heat map of correlation analysis between bacterial genus (a) and fungal genus (b) and sediment physical and chemical indicators($r<0.001$, "***", $r<0.01$, "**", $r<0.05$, "*", the 20 species with the most significant correlation coefficients).**

## Discussion

### Microbial community diversity and distribution

In the analysis at the bacterial phylum level, the results showed that the dominant phyla in the Erhai Lake sediments included *Proteobacteria*, *Bacteroidetes*, *Chloroflexi*, *Acidobacteria*, and *Actinobacteria*. These dominant phyla are common in plateau lake sediments, consistent with previous research findings Pan, Lin [13–15]. In particular, *Proteobacteria* was the most abundant phylum in all sediment samples, aligning with the general dominance of *Proteobacteria* in plateau lake sediments [15]. *Proteobacteria*, *Bacteroidetes*, and *Acidobacteria* play key roles in important biochemical reactions such as anaerobic ammonium oxidation and sulfate reduction, thus maintaining ecosystem functions under various environmental conditions [16,17]. Notably, on the west shore, the high abundance of *Proteobacteria* and *Bacteroidetes* may reflect higher nutrient levels and pollution loads in this area.

Further analysis found that the abundance of *Proteobacteria* and *Bacteroidetes* in the sediments of the west shore was significantly higher than that of the east and north shores, indicating more severe environmental pollution on the west shore. This reflects the selective pressure of environmental conditions in different regions on the structure of microbial

communities [18]. For instance, the higher abundance of *Proteobacteria* on the west shore may be related to the higher organic matter content and degradation activity in the region, consistent with the high abundance of *Proteobacteria* in eutrophic lakes reported in previous studies [19]. Additionally, the microbial community structure in the sediments of the west shore is relatively stable and consistent, whereas that of the east and north shores is more scattered. This difference may be attributed to ecological restoration measures on the west shore, which help improve sediment quality and promote microbial diversity and stability. Erhai Lake completed the ecological restoration project on the west shore in 2020. Studies have shown that *Proteobacteria*, *Bacteroidetes*, *Chloroflexi*, and other bacterial phyla dominate in the sediments of plateau lakes [14]. These dominant phyla play a key role in material cycling and serve as indicators of environmental change [20]. Ecological restoration measures such as vegetation restoration and pollutant removal can significantly alter the structure and function of microbial communities, thereby enhancing ecosystem stability and resilience [21], and positively affecting ecological restoration. In the genus level analysis, the relative abundance of *Luteolibacter* sp. and Sva0081_sediment_group in the sediments of the west shore was significantly higher than that of the east and north shores, while the relative abundance of *Thiobacillus* sp. and *Ellin6067* sp. in the sediments of the north shore was higher than in other areas. *Thiobacillus* sp. can effectively remove nitrate nitrogen from the water body and significantly influence the transformation and degradation of sulfur, enhancing the self-purification capacity of the water body [22,23]. This finding suggests that the north shore may have a higher self-purification capacity, which helps maintain water quality.

In terms of fungi, research results show that the richness of fungal communities in Erhai Lake sediments is high, with dominant genera including *Frontonia* sp., *Prorodon* sp., *Arcuospathidium* sp., *Emericellopsis* sp., *Cyrtohymena* sp., and *Vorticellides* sp [8]. The relative abundance of *Frontonia* sp.and *Prorodon* sp. in sediments on the west shore was significantly higher than in other areas, indicating that ecological restoration measures have a significant impact on fungal community structure. The relative abundance of *Arcuospathidium* sp. and *Cyrtohymena* sp. in the north shore sediments is higher than in other areas, suggesting that these fungi may have stronger adaptability to the natural environmental conditions of the north shore. The relative abundance of *Emericellopsis* sp. was highest in sediments on the east shore, indicating that the environment on the east shore may be more suitable for the growth and reproduction of these fungi. The study also found that genera such as *Gibellulopsis* sp., *Gomphonema* sp., *Cytospora* sp., *Tintinnidium* sp., *Gaertneriomyces* sp. and *Trichoderma* sp. have extremely significant negative or positive correlations with pH and Eh, highlighting the significant impact of these environmental factors on fungal community structure [4].

## Effects of environmental factors on microbial communities

First, from the perspective of environmental factors, the pH value of Erhai Lake is alkaline as a whole. The pH value on the shore is significantly higher than that on the west shore and the north shore, and the pH value on the west shore is the lowest. pH is a key environmental factor affecting the structure of bacterial and fungal communities [24]. A higher pH value has an inhibitory effect on some microorganisms and promotes the reproduction of alkali-resistant microorganisms, such as *Crenothrix* sp.and *Thiobacillus* sp. [4]. This explains why the diversity of bacteria and fungi in the sediments of the east shore is low.

In terms of nutrients, the contents of TN, TP, TOC, and $NH_3$-N in the sediments of the east shore were all low, while the contents of these nutrients in the sediments of the west shore and the north shore were higher. The rich organic matter and nitrogen and phosphorus compounds provide sufficient nutrients for the growth of microorganisms, which is consistent with the findings of Shen, Li [25]. The north and west shore areas may be more polluted by nitrogen sources, such as agricultural runoff and domestic sewage discharge. Additionally, the north shore has the longest water system and more agricultural activities in the surrounding area, resulting in more agricultural pollutants entering the lake. This is consistent with the highest TN and TOC contents in the sediments of the north shore observed in this study. The river system on the east shore is the shortest and there is less agricultural pollution, so the TN, TP, and TOC contents in its sediments are the lowest. This further demonstrates that the length of the river system and the surrounding agricultural

activities have an important influence on the physicochemical properties of the sediments and the structure of the microbial community. This can also be evidenced by the abundance of microorganisms. The relative abundance of nitrogen cycle-related microorganisms, such as *Thiobacillus* sp., is higher on the north shore, indicating that nitrogen cycle activity in this area is strong, which is conducive to the natural purification of nitrogen pollution. In contrast, the lower TN content on the east shore may have limited the growth of these microorganisms. These results indicate that the pollution levels of nitrogen, phosphorus, and organic carbon in the sediments of Erhai Lake are high. High concentrations of organic matter and nutrients promote the activity of microorganisms in sediments, leading to changes in the structure of microbial communities [13,15].

The heat map and Bray-Curtis distance analysis results showed that the microbial community structure between different sampling points in Erhai Lake was significantly different. In particular, the sampling sites on the west shore have more similar microbial community structures due to ecological restoration, while the microbial community structures on the east and north shores are relatively dispersed. This phenomenon is consistent with previous studies, as ecological restoration can significantly affect the composition and structure of microbial communities by improving environmental conditions [26,27]. This phenomenon was further verified in NMDS plots and ANOSIM analysis. The results showed that there was no significant difference in the microbial community structure between the east shore and the north shore, while the difference between the west shore and the east shore and the north shore was significant. The impact of ecological restoration was fully reflected in these analyses, showing that restoration measures effectively changed the microbial community structure of the west shore. Ecological restoration measures may include vegetation restoration, pollutant removal, etc. These measures can increase soil organic matter content and microbial activity, thereby stabilizing the microbial community.

Further Spearman correlation analysis found that pH, Eh, $NH_3$-N, MC, and TOC are key factors affecting the fungal community structure. In the bacterial community, bacterial genera such as *Terrimonas* sp., *Lacibacter* sp. and *Truepera* sp. were significantly negatively correlated with TOC, while *Thermoanaerobaculum* sp., ADurb.Bin063−1, and other bacterial genera were significantly positively correlated with TOC. This indicates differences in the adaptability of different bacterial genera to organic carbon content [28]. In the fungal community, *Gibellulopsis* sp., *Gomphonema* sp., *Cytospora* sp., and other fungal genera were significantly negatively correlated with pH and Eh, indicating that these genera grow better in environments with lower pH and higher conductivity.

Redundancy analysis (RDA) and canonical correspondence analysis (CCA) further verified the above results. RDA results showed that TN, MC, and TOC were the main environmental factors affecting the structure of bacterial communities, while the importance of pH and TP on different axes indicated that these factors had varying effects on the structure of microbial communities under different environmental conditions. CCA results showed that pH, Eh, and MC were the main environmental factors affecting the structure of fungal communities, which was consistent with the results of the Spearman correlation analysis.

### Study limitations

While this study provides novel insights into spatial heterogeneity of sediment microbial communities in Erhai Lake, several limitations merit consideration for contextualizing the findings and guiding future work.

Sampling was conducted during a single hydrological period (early wet season, June 2020), which effectively captured spatial heterogeneity driven by shoreline-specific anthropogenic impacts (agricultural inputs on northern/western shores; minimal disturbance on eastern shore). However it does not account for potential seasonal dynamics [4,9]. For instance, microbial communities in lake sediments may fluctuate with wet/dry seasons due to changes in nutrient loading, temperature, and hydrology [5,15]. Future studies should incorporate multi-season sampling to resolve temporal patterns. Nevertheless, our findings provide a robust baseline for understanding spatial drivers, which dominate microbial structuring in anthropogenically impacted lakes [18].

While RDA and Spearman analyses robustly identified associations between environmental factors (pH, TN, TOC) and microbial taxa, these correlations do not imply direct causation. And *Thiobacillus* sp. enrichment in nitrogen-rich northern sediments (Fig 3a) aligns with its known role in sulfur-driven denitrification [22], yet functional causality requires validation. The negative correlation between *Lacibacter* and TOC (Fig 9a) may reflect niche competitionrather than TOC causing *Lacibacter* decrease. The correlation analyses (RDA/Spearman) effectively screened key environmental drivers (e.g., pH, TOC), they cannot disentangle causal mechanisms due to ecosystem complexity. Future studies should employ microcosm experiments (Sediment incubations with controlled factor manipulation) to validate causal drivers, and metagenomic profiling to resolve functional linkages.

Taxonomic shifts imply functional adaptations to environmental stressors. For instance, high *Proteobacteria*/*Bacteroidetes* abundance on the western shore (Fig 2a) aligns with their capacity for polysaccharide degradation [16], consistent with elevated TOC loads. *Thiobacillus* sp. enrichment in nitrogen-rich northern sediments (Fig 3a) correlates with its known role in sulfur-driven denitrification [22], potentially enhancing N-removal. Ecological restoration on the western shore coincided with stable fungal communities (*Frontonia* sp., *Prorodon* sp.; Fig 3b), which may promote organic matter decomposition resilience [8]. While phylogeny often predicts function, this approach cannot resolve strain-level metabolic capabilities or biochemical pathways. These patterns support hypothesized biogeochemical roles, future metagenomic profiling is needed to resolve functional genes and the direct links between community stability and ecosystem function require further validation.

Heavy metals and hydrodynamic condition-known influencers of microbial communities-were not measured. However, their impacts in Erhai Lake are likely secondary to the dominant nutrient/pH gradients identified here. Low sediment metal concentrations in Erhai (Cd, 0.10–2.00 mg/kg; Zn, 30–170 mg/kg) [29,30], below thresholds causing significant microbial shifts (Cd, 8.8 mg/kg; Zn, 300 mg/kg) [31]. In freshwater sediments, organic carbon and nitrogen availability consistently emerge as primary drivers of microbial assembly, whereas trace metals exert secondary effects unless present at acutely toxic levels [17,18]. The strong spatial covariation observed between microbial clusters (e.g., *Proteobacteria*/*Bacteroidetes*-TOC; *Thiobacillus*-TN) and anthropogenic nutrient inputs (agricultural runoff on northern/western shores) aligns with known microbial responses to eutrophication [15], suggesting nutrients override other factors. Although heavy metals occur in Erhai sediments, their concentrations are comparatively low and exhibit weaker correlations with TOC in regional lakes [29]. Hoever, these factors may fine tune community assembly and, under long-term accumulation, affect community structure. In future research, it is necessary to pay attention to their direct and indirect effects in order to obtain a more comprehensive trend of microbial community structure changes.

## Conclusion and outlook

This study systematically explored the diversity of microbial communities in the surface sediments of Erhai Lake and its relationship with environmental factors through multiple analysis methods. The results show that the microbial community structure in the surface sediments of Erhai Lake is significantly affected by various environmental factors. The sediments on the north shore have the highest nutrient content, but the bacterial diversity is lower and the fungal diversity is the lowest, which may be due to agricultural pollution. The natural environmental conditions on the east shore are better, and the bacterial diversity and specificity are the highest, indicating that less human interference is conducive to microbial diversity. The ecological restoration project on the west shore increased the input of organic matter and promoted fungal diversity and specificity, while bacterial diversity was relatively low; however, the microbial community structure was relatively stable and similar. *Proteobacteria* was the dominant bacterial phylum at all sampling points, indicating its important position in plateau lake sediments. Environmental factors such as pH, Eh, TN, MC, and TOC have significant effects on the microbial community structure of Erhai Lake sediments.

Despite these insights, our conclusions are tempered by limitations, such as single-time sampling restricting extrapolation to seasonal dynamics, correlation analyses (RDA/Spearman) not establishing causality, taxonomic-based inferences

lack functional validation and unmeasured factors (e.g., trace metals, hydrodynamics) contributing residual variance. Future work should integrate multi-season sampling, omics-based functional profiling, and expanded environmental variables to resolve mechanistic links between community structure and ecosystem function.

**Permits:** All field sampling activities were formally authorized as part of the completion environmental protection acceptance for the "Construction Project for Ecological Restoration and Wetland Construction of the Lakeside Buffer Zone around Erhai Lake in Dali City". This program was commissioned by the Dali City Erhai Lake Administration Bureau, the governmental authority responsible for managing Erhai Lake.

## Supporting information

**S1 Dataset.**

(DOCX)

## Author contributions

**Conceptualization:** Hao-Ran Bao.

**Funding acquisition:** Xiao-Long Sun.

**Supervision:** Jia-Zhuo Qu, Zhe-Xi Luan.

**Writing – original draft:** Cong-gao Yang, Hao-Qin Xiong.

**Writing – review & editing:** Hong Xiang, Xiao-Long Sun, Li Zhen.

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
