## [Decision Letter · Decision Letter 0]

17 Jun 2025

PONE-D-25-17024Spatial Heterogeneity of Microbial Community Structure and Its Environmental Drivers in Surface Sediments of Erhai LakePLOS ONE

Dear Dr.  Sun,

Thank you for submitting your manuscript to PLOS ONE. After careful consideration, we feel that it has merit but does not fully meet PLOS ONE’s publication criteria as it currently stands. Therefore, we invite you to submit a revised version of the manuscript that addresses the points raised during the review process.

We look forward to receiving your revised manuscript.

Kind regards,

Barathan Balaji Prasath

Academic Editor

PLOS ONE

Journal Requirements:

“This study was sponsored by the Yunnan Province Agricultural Basic Research Joint Special Project (202301BD070001-065), the Key R&D Plan Projects in Yunnan Province(202203AC100002-03) and the Scientific Research Fund of Yunnan Provincial Education Department, China (2022J0524).”

5. We note that your Data Availability Statement is currently as follows: All relevant data are within the manuscript and in Supporting Information files.

6. We note that Figure 1 in your submission contain map/satellite images which may be copyrighted. All PLOS content is published under the Creative Commons Attribution License (CC BY 4.0), which means that the manuscript, images, and Supporting Information files will be freely available online, and any third party is permitted to access, download, copy, distribute, and use these materials in any way, even commercially, with proper attribution. For these reasons, we cannot publish previously copyrighted maps or satellite images created using proprietary data, such as Google software (Google Maps, Street View, and Earth). For more information, see our copyright guidelines: http://journals.plos.org/plosone/s/licenses-and-copyright.

Reviewers' comments:

Reviewer's Responses to Questions

**Comments to the Author**

1. Is the manuscript technically sound, and do the data support the conclusions?

Reviewer #1: Yes

Reviewer #2: Yes

2. Has the statistical analysis been performed appropriately and rigorously? 

Reviewer #1: Yes

Reviewer #2: Yes

3. Have the authors made all data underlying the findings in their manuscript fully available?

Reviewer #1: Yes

Reviewer #2: Yes

4. Is the manuscript presented in an intelligible fashion and written in standard English?

Reviewer #1: Yes

Reviewer #2: Yes

5. Review Comments to the Author

Reviewer #1: Specific comments:

Despite providing valuable insights into the spatial dynamics of microbial communities in Erhai Lake sediments, this study has several limitations:

1- It was based on surface sediment samples collected at a single time point, which may not translate the seasonal or temporal fluctuations in microbial communities and environmental factors.

2- The study primarily depends on correlation analyses (RDA and Spearman), which identify associations but don’t determine the reason behind that.

3- Functional insights into microbial roles were limited based on taxonomic composition rather than direct metagenomic or metatranscriptomic analyses, limiting conclusions about ecosystem function.

4- The study did not assess the potential influence of additional environmental factors (e.g., heavy metals, microplastics, or hydrodynamic conditions) that may also impact microbial distribution.

Future studies incorporating temporal sampling and functional assays would enhance understanding of microbial contributions to lake ecosystem health and resilience, so please consider adding a paragraph discussing the limitations of the study before the conclusion.

Reviewer #2: Line 72 – km2 (Superscript to be used)

Line 73 – m3 (Superscript to be used)

Line 83 – 95 Use Past tense to describe the collection process.

Line 98 – Change the title to Determination of Physical and Chemical Properties of Sediment

Line 281 to Line 303 – Use sp. after the genus names. (Change in discussion part too)

Line 372 – Change to italics

6. PLOS authors have the option to publish the peer review history of their article (what does this mean? ). If published, this will include your full peer review and any attached files.

**Do you want your identity to be public for this peer review?** For information about this choice, including consent withdrawal, please see our Privacy Policy .

Reviewer #1: No

Reviewer #2: No

---

## [Author Response · Author response to Decision Letter 1]

8 Jul 2025

29-June-2025

Dear reviewers,

We would like to thank you for the careful and constructive reviews. we revised the manuscript entitled “Spatial heterogeneity of microbial community structure and its environmental drivers in surface sediments of erhai lake”(Manuscript Number: PONE-D-25-17024) according to your suggestions. Corrections to manuscripts have been made in red font. We have uploaded a clean version with the latest changes.

The following is a response to the reviewer’s comments:

Reviewer #1:

Despite providing valuable insights into the spatial dynamics of microbial communities in Erhai Lake sediments, this study has several limitations:

1- It was based on surface sediment samples collected at a single time point, which may not translate the seasonal or temporal fluctuations in microbial communities and environmental factors.

2- The study primarily depends on correlation analyses (RDA and Spearman), which identify associations but don’t determine the reason behind that.

3- Functional insights into microbial roles were limited based on taxonomic composition rather than direct metagenomic or metatranscriptomic analyses, limiting conclusions about ecosystem function.

4- The study did not assess the potential influence of additional environmental factors (e.g., heavy metals, microplastics, or hydrodynamic conditions) that may also impact microbial distribution.

Future studies incorporating temporal sampling and functional assays would enhance understanding of microbial contributions to lake ecosystem health and resilience, so please consider adding a paragraph discussing the limitations of the study before the conclusion.

Response: Thank you for your opinion. We sincerely appreciate the valuable feedback, which has significantly improved the rigor and completeness of our study. We have carefully addressed all raised limitations by adding a dedicated "Study limitations" section (Section 4.3, Lines 427-476) in the Discussion and expanding the Conclusion and outlook section (Section 5, Lines 492-498).

Specifically, we made the following changes:

While this study provides novel insights into spatial heterogeneity of sediment microbial communities in Erhai Lake, several limitations merit consideration for contextualizing the findings and guiding future work.

Sampling was conducted during a single hydrological period (early wet season, June 2020), which effectively captured spatial heterogeneity driven by shoreline-specific anthropogenic impacts (agricultural inputs on northern/western shores; minimal disturbance on eastern shore). However it does not account for potential seasonal dynamics[4, 9]. For instance, microbial communities in lake sediments may fluctuate with wet/dry seasons due to changes in nutrient loading, temperature, and hydrology [5, 16]. Future studies should incorporate multi-season sampling to resolve temporal patterns. Nevertheless, our findings provide a robust baseline for understanding spatial drivers, which dominate microbial structuring in anthropogenically impacted lakes[19].

While RDA and Spearman analyses robustly identified associations between environmental factors (pH, TN, TOC) and microbial taxa, these correlations do not imply direct causation. And Thiobacillus sp. enrichment in nitrogen-rich northern sediments (Fig 3(a)) aligns with its known role in sulfur-driven denitrification[23], yet functional causality requires validation. The negative correlation between Lacibacter and TOC (Fig 9(a)) may reflect niche competitionrather than TOC causing Lacibacter decrease. The correlation analyses (RDA/Spearman) effectively screened key environmental drivers (e.g., pH, TOC), they cannot disentangle causal mechanisms due to ecosystem complexity. Future studies should employ microcosm experiments (Sediment incubations with controlled factor manipulation) to validate causal drivers, and metagenomic profiling to resolve functional linkages.

Taxonomic shifts imply functional adaptations to environmental stressors. For instance, high Proteobacteria /Bacteroidetes abundance on the western shore (Fig 2(a)) aligns with their capacity for polysaccharide degradation [17], consistent with elevated TOC loads. Thiobacillus sp. enrichment in nitrogen-rich northern sediments (Fig 3(a)) correlates with its known role in sulfur-driven denitrification [23], potentially enhancing N-removal. Ecological restoration on the western shore coincided with stable fungal communities (Frontonia sp., Prorodon sp.; Fig 3(b)), which may promote organic matter decomposition resilience [8]. While phylogeny often predicts function, this approach cannot resolve strain-level metabolic capabilities or biochemical pathways. These patterns support hypothesized biogeochemical roles, future metagenomic profiling is needed to resolve functional genes and the direct links between community stability and ecosystem function require further validation.

Heavy metals and hydrodynamic condition-known influencers of microbial communities-were not measured. However, their impacts in Erhai Lake are likely secondary to the dominant nutrient/pH gradients identified here. Low sediment metal concentrations in Erhai (Cd, 0.10–2.00mg/kg; Zn, 30–170 mg/kg) [29, 30], below thresholds causing significant microbial shifts (Cd, 8.8 mg/kg; Zn, 300 mg/kg)[31]. In freshwater sediments, organic carbon and nitrogen availability consistently emerge as primary drivers of microbial assembly, whereas trace metals exert secondary effects unless present at acutely toxic levels [18, 19]. The strong spatial covariation observed between microbial clusters (e.g., Proteobacteria/Bacteroidetes-TOC; Thiobacillus-TN) and anthropogenic nutrient inputs (agricultural runoff on northern/western shores) aligns with known microbial responses to eutrophication [16], suggesting nutrients override other factors. Although heavy metals occur in Erhai sediments, their concentrations are comparatively low and exhibit weaker correlations with TOC in regional lakes [29] . Hoever, these factors may fine tune community assembly and, under long-term accumulation, affect community structure. In future research, it is necessary to pay attention to their direct and indirect effects in order to obtain a more comprehensive trend of microbial community structure changes.

To align with the limitations discussed, we strengthened the conclusion’s forward-looking statements (Section 5, Lines 492–498).

Despite these insights, our conclusions are tempered by limitations, such as single-time sampling restricting extrapolation to seasonal dynamics, correlation analyses (RDA/Spearman) not establishing causality, taxonomic-based inferences lack functional validation and unmeasured factors (e.g., trace metals, hydrodynamics) contributing residual variance. Future work should integrate multi-season sampling, omics-based functional profiling, and expanded environmental variables to resolve mechanistic links between community structure and ecosystem function.

Reviewer #2:

1.Line 72 – km2 (Superscript to be used)

2.Line 73 – m3 (Superscript to be used)

3.Line 83 – 95 Use Past tense to describe the collection process.

4.Line 98 – Change the title to Determination of Physical and Chemical Properties of Sediment

5.Line 281 to Line 303 – Use sp. after the genus names. (Change in discussion part too)

6.Line 372 – Change to italics

Response: We appreciate your attention to technical details and have implemented all suggested revisions as follows:

Revision:

1.Line 74: 250 km2.

2.Line 75: 2.88 billion m3.

3.All procedural descriptions in Section 2.1 (Lines 87–98) have been converted to past tense.

Sediment samples (0–10 cm depth) were collected from designated sampling points on the eastern, northern, and western shores of Erhai Lake using a grab sampler. Samples from each point were transferred to a clean plastic bucket and thoroughly homogenized by hand (using gloved hands) to eliminate vertical stratification effects. From each homogenized composite, two subsamples were allocated, one for physicochemical analysis and another for microbiological analysis. Both subsamples were placed in separate ziplock bags labeled with their respective sampling point codes and immediately stored in a low-temperature incubator. Triplicate samples were collected per sampling point. Following collection, subsamples designated for microbiological analysis were transported to an accredited testing facility. The remaining subsamples were air-dried at ambient temperature in a ventilated environment. During drying, macroscopic debris (plant matter, animal remains, and gravel) was manually removed. Processed samples were then sealed in airtight bags for subsequent physicochemical characterization.

4. Section 2.2 title changed to:"Determination of physical and chemical properties of sediment"(Line 101)

5.All references to bacterial/fungal genera (where species-level identification was not resolved) now include "sp.":

Seven bacteria (Terrimonas sp., Lacibacter sp., Truepera sp., Meiothermus sp., mle1-7, JGI_0001001-H03, and Calothrix sp.KVSF5) showed a significant negative correlation with TOC (P<0.01), while three bacteria (Thermoanaerobaculum sp., ADurb.Bin063-1 and Syntrophorhabdus sp.) had a significant positive correlation with TOC. Five bacteria (Terrisporobacter sp., Lacibacter sp., Truepera sp., mle1-7, and Meiothermus sp.) displayed a significant positive correlation with pH, whereas three bacteria (Methyloversatilis sp., Geobacter sp., and Syntrophorhabdus sp.) had a significant negative correlation with pH. For Eh, the same five bacteria showed a significant negative correlation, while Methyloversatilis sp., Geobacter sp., and Syntrophorhabdus sp. had a significant positive correlation. Regarding NH3-N, five bacteria (Terrisporobacter sp., Lacibacter sp., Truepera sp., Meiothermus sp., and JGI_0001001-H03) showed a significant negative correlation, whereas Subgroup_23, BD1-7_clade, and ADurb.Bin063-1 displayed a significant positive correlation. Lastly, six bacteria (Sulfurifustis sp., Pleurocapsa sp.PCC-7319, Lacibacter sp., Truepera sp., Meiothermus sp., and JGI_0001001-H03) had a significant negative correlation with MC, while Thermoanaerobaculum sp., ADurb.Bin063-1, and Syntrophorhabdus sp. exhibited a significant positive correlation with MC.

Six fungal genera (Gibellulopsis sp., Gomphonema sp., Cytospora sp., Tintinnidium sp., Gaertneriomyces sp. and Trichoderma sp.) showed a significant negative correlation with pH (P<0.01). Seven genera (Gibellulopsis sp., Gomphonema sp., Cytospora sp., Paramecium sp., Tintinnidium sp., Gaertneriomyces sp. and Trichoderma sp.) were significantly positively correlated with Eh (P<0.01). Five genera (Askenasia sp., Fusarium sp., Gomphonema sp., Tintinnidium sp. and Gibellulopsis sp.) exhibited a significant positive correlation with NH3-N (P<0.01). Cytospora sp., Gomphonema sp., Cladosporium sp. and Leptodiscella sp. displayed a significant positive correlation with TOC (P<0.05). Three genera (Fusarium sp., Leptodiscella sp. and Cytospora sp.) were significantly positively correlated with NO3--N (P<0.05). Cytospora sp.and Gomphonema sp.also showed a significant positive correlation with MC (P<0.05). Similar to bacteria, TN, TP, and DOC had little impact on the structure of fungal communities in sediments.

(Applied consistently in Section 3.2.2, 3.2.3, 4.1, and 4.2)

Exception: Higher taxonomic ranks (e.g., phyla like Proteobacteria) remain unchanged.

6.Line 374: such as Crenothrix sp.and Thiobacillus sp.

We sincerely hope that these revisions can make the manuscript more perfect to meet your requirements. Thank you again for your review and guidance. All the revised contents have been clearly marked in red in the manuscript for your review.

Sincerely,

Xiaolong Sun,

Professor, Southwest Forestry University

Kunming 650224, Yunnan Province, China

sunxl96@gmail.com

---

## [Decision Letter · Decision Letter 1]

23 Jul 2025

Spatial Heterogeneity of Microbial Community Structure and Its Environmental Drivers in Surface Sediments of Erhai Lake

PONE-D-25-17024R1

Dear Dr. Xiaolong Sun,

We’re pleased to inform you that your manuscript has been judged scientifically suitable for publication and will be formally accepted for publication once it meets all outstanding technical requirements.

Kind regards,

Barathan Balaji Prasath

Academic Editor

PLOS ONE

Additional Editor Comments (optional):

Reviewers' comments:

Reviewer's Responses to Questions

**Comments to the Author**

1. If the authors have adequately addressed your comments raised in a previous round of review and you feel that this manuscript is now acceptable for publication, you may indicate that here to bypass the “Comments to the Author” section, enter your conflict of interest statement in the “Confidential to Editor” section, and submit your "Accept" recommendation.

Reviewer #1: All comments have been addressed

Reviewer #2: All comments have been addressed

2. Is the manuscript technically sound, and do the data support the conclusions?

Reviewer #1: Yes

Reviewer #2: Yes

3. Has the statistical analysis been performed appropriately and rigorously? 

Reviewer #1: Yes

Reviewer #2: Yes

4. Have the authors made all data underlying the findings in their manuscript fully available?

Reviewer #1: Yes

Reviewer #2: Yes

5. Is the manuscript presented in an intelligible fashion and written in standard English?

Reviewer #1: Yes

Reviewer #2: Yes

6. Review Comments to the Author

Reviewer #1: Dear authors,

All comments have been addressed. I have no more comments for the manuscript. Good work overall.

Reviewer #2: Methodology is detailed and well written. Technical errors have been effectively changed and other minor revisions are carried out well

7. PLOS authors have the option to publish the peer review history of their article (what does this mean? ). If published, this will include your full peer review and any attached files.

**Do you want your identity to be public for this peer review?** For information about this choice, including consent withdrawal, please see our Privacy Policy .

Reviewer #1: No

Reviewer #2: **Yes: ** Jeyanthi Selvakumaran

---

## [Editor Report · Acceptance letter]

PONE-D-25-17024R1

PLOS ONE

Dear Dr. Sun,

I'm pleased to inform you that your manuscript has been deemed suitable for publication in PLOS ONE. Congratulations! Your manuscript is now being handed over to our production team.

Kind regards,

on behalf of

Dr. Barathan Balaji Prasath

Academic Editor

PLOS ONE